# Alternate subunit assembly diversifies the function of a bacterial toxin

Casey C. Fowler [1,2], Gabrielle Stack[1], Xuyao Jiao[1], Maria Lara-Tejero[1] & Jorge E. Galán [1]

Bacterial toxins with an AB$_5$ architecture consist of an active (A) subunit inserted into a ring-like platform comprised of five delivery (B) subunits. *Salmonella* Typhi, the cause of typhoid fever, produces an unusual A$_2$B$_5$ toxin known as typhoid toxin. Here, we report that upon infection of human cells, *S*. Typhi produces two forms of typhoid toxin that have distinct delivery components but share common active subunits. The two typhoid toxins exhibit different trafficking properties, elicit different effects when administered to laboratory animals, and are expressed using different regulatory mechanisms and in response to distinct metabolic cues. Collectively, these results indicate that the evolution of two typhoid toxin variants has conferred functional versatility to this virulence factor. More broadly, this study reveals a new paradigm in toxin biology and suggests that the evolutionary expansion of AB$_5$ toxins was likely fueled by the plasticity inherent to their structural design coupled to the functional versatility afforded by the combination of homologous toxin components.

[1] Department of Microbial Pathogenesis, Yale University School of Medicine, New Haven, CT 06536, USA. [2]Present address: Department of Biological Sciences, University of Alberta, Edmonton, Alberta T6G 2E9, Canada. Correspondence and requests for materials should be addressed to J.E.G. (email: jorge.galan@yale.edu)

A$B_5$ toxins are a structurally similar but functionally diverse class of virulence factors that are widespread in bacteria. They play an important and well-documented role in the pathogenesis of many pathogens of great public health importance including *Bordetella pertussis*, *E. coli*, *Vibrio cholerae*, *Shigella* spp. and *Salmonella* spp[1,2]. Their name is derived from their architectural organization, which consists of a pentameric "B" subunit, which targets the toxin to specific cells, and an associated catalytic "A" subunit, which is responsible for the harmful effects on cellular targets. *Salmonella enterica* serovar Typhi (*S.* Typhi) is the etiological agent of typhoid fever, a major global health problem[3,4]. An assortment of evidence indicates that typhoid toxin is responsible for some of the more severe symptoms of typhoid fever[5–7]. Compared to other AB-type toxins, typhoid toxin is highly unusual in that two A subunits, CdtB, a DNAse, and PltA, an ADP-ribosyltransferase, associate with a single pentameric B subunit, PltB, resulting in a unique $A_2B_5$ architecture[6]. This unusual composition appears to be the result of typhoid toxin's remarkable evolutionary history, during which two classes of AB-type toxins – cytolethal distending toxins (CDTs) and pertussis-like toxins - amalgamated to produce a single toxin[8]. A structural and biochemical investigation into the evolution of typhoid toxin revealed that this class of toxins exhibits remarkable plasticity in that heterologous co-expression of various combinations of homologs from other bacteria produced active typhoid-toxin-like complexes[8]. These observations raise the question whether alternative forms of $AB_5$ toxins could be assembled from homologous subunits encoded by the same bacterium.

Here we show that in *S.* Typhi this is the case and that an alternative form of typhoid toxin is assembled involving a B subunit homolog encoded elsewhere in its chromosome. We also show that the alternative form of typhoid toxin exhibits different biological properties and that the expression of its B subunit is controlled by a different regulatory network in response to different metabolic cues. These results have important implications for the biology of typhoid toxin and $AB_5$ toxins in general.

## Results

**Many salmonellae encode $AB_5$ toxins and "orphan" B subunits.** We previously observed that, surprisingly, orthologous $AB_5$ toxin components encoded by different salmonellae are able to assemble into functional toxins[8]. This suggested to us that this might be an evolutionarily-conserved phenomenon exploited by *Salmonella* to produce diversified toxins. To test this hypothesis, we searched the NCBI genome database for salmonellae that encode both typhoid toxin (i.e. *pltB*, *pltA* and *cdtB*) as well as additional putative toxin-encoding genes homologous to typhoid toxin components. In agreement with previous reports[9,10], we found that the typhoid toxin islet is found in an assortment of *Salmonella* lineages and has a distribution that is consistent with having been transferred horizontally within the genus in multiple independent events. We found that, in addition to the core toxin locus, almost every typhoid toxin-encoding strain that we identified encoded a second homolog of the *pltB* delivery subunit (Supplementary Fig. 1). Remarkably, the genomic context of this putative second delivery subunit varies considerably among salmonellae. In some lineages, such as the *arizonae* and *diarizonae* subspecies, the additional *pltB* homolog is encoded immediately upstream of *pltB*, while in most cases, including the Typhi and Paratyphi serovars, it is found at a distant genome location as an "orphan" B subunit (Fig. 1a, Supplementary Fig. 1). Importantly, although the majority of sequenced *Salmonella* strains do not encode typhoid toxin, we were unable to identify any strains that encode an orphan B subunit in the absence of typhoid toxin. The remarkable co-occurrence of these two genetic elements

across different branches of the *Salmonella* genus supports the hypothesis that *Salmonella* may assemble alternative forms of typhoid toxins from orthologous components.

**S. Typhi produces two distinct forms of typhoid toxin.** As alluded above, the *S.* Typhi genome contains a gene that encodes a polypeptide that shares 28% amino acid sequence identity with PltB at a locus distant from the typhoid toxin islet (Fig. 1a, Supplementary Figs. 1–2)[10]. This gene, *sty1364*, has been renamed *pltC* in accordance with the findings presented below. A thoroughly degraded ADP-ribosyltransferase pseudo-gene, *sty1362*, resides immediately upstream of *pltC*, suggesting that this locus encoded a complete $AB_5$-type toxin at some point in its evolutionary history (Supplementary Fig. 1). We hypothesized that PltC may associate with PltA and CdtB to produce an alternative form of typhoid toxin (Fig. 1a). In testing this hypothesis, we found that *pltC* exhibited a similar pattern of expression to the genes encoding other components of typhoid toxin[5,11,12] in that, although undetectable under standard laboratory conditions, it was induced >100-fold following *S.* Typhi infection of cultured human epithelial cells (Fig. 1b). Similarly, *pltC* expression was strongly induced in *S. Typhi* grown in TTIM, a growth medium that mimics some aspects of the intracellular environment and is permissive for typhoid toxin expression *in vitro*[12] (Fig. 1c). Using affinity chromatography coupled to LC–MS/MS and western blot analysis we readily detected an interaction between PltC and both CdtB and PltA after *S.* Typhi growth in TTIM medium as well as after infection of cultured epithelial cells (Fig. 1d, e, Supplementary Data 1). Importantly, PltC did not interact with CdtB in the absence of PltA, indicating that similar to the holotoxin assembled with PltB, PltC forms a complex with CdtB only through its interaction with PltA (Fig. 1d, e). Although most $AB_5$ toxins employ a homo-pentameric delivery platform[2], pertussis toxin is comprised of a heteropentameric delivery platform assembled from different but structurally related B subunits[13,14]. Because PltB is not efficiently detected using our LC–MS/MS protocol even in purified toxin preparations, we were unable to determine whether PltB and PltC form heteromeric delivery platforms using this approach. Therefore, to explore this issue we employed immunoprecipitation coupled with western blot analysis using an anti-PltB antibody in *S.* Typhi-infected cells and in *S.* Typhi grown in TTIM (Fig. 1e, f). Under both of these conditions, PltB was readily detected in samples affinity purified from a tag present in CdtB, but was undetectable in parallel samples purified from a tag present in PltC. These results indicate that, most likely, typhoid toxin does not exhibit a single heteromeric B subunit architecture but rather it is assembled in two alternative configurations with PltB or PltC as its homopentameric subunit (Fig. 1e, f). We also observed an increase in the amount of PltB that co-immunoprecipitated with CdtB in a Δ*pltC* strain compared to wild type, indicating that, in the absence of PltC, more PltB-containing toxin is assembled thus suggesting that the two delivery subunits compete for their association to the active subunits (Fig. 1f, Supplementary Fig. 3). Following growth in TTIM, there also appears to be more PltB in the whole cell lysates in the Δ*pltC* mutant strain compared to wild type, which might indicate that free B subunits are degraded more readily than those incorporated into the toxin (Fig. 1f). Collectively, these data indicate that upon infection of human cells *S.* Typhi assembles two distinct typhoid toxins with the same enzymatic "A" subunits but distinct delivery platforms or "B" subunits (Fig. 1a).

**The two typhoid toxins exhibit significant functional differences.** To assess the function of the typhoid toxin assembled with

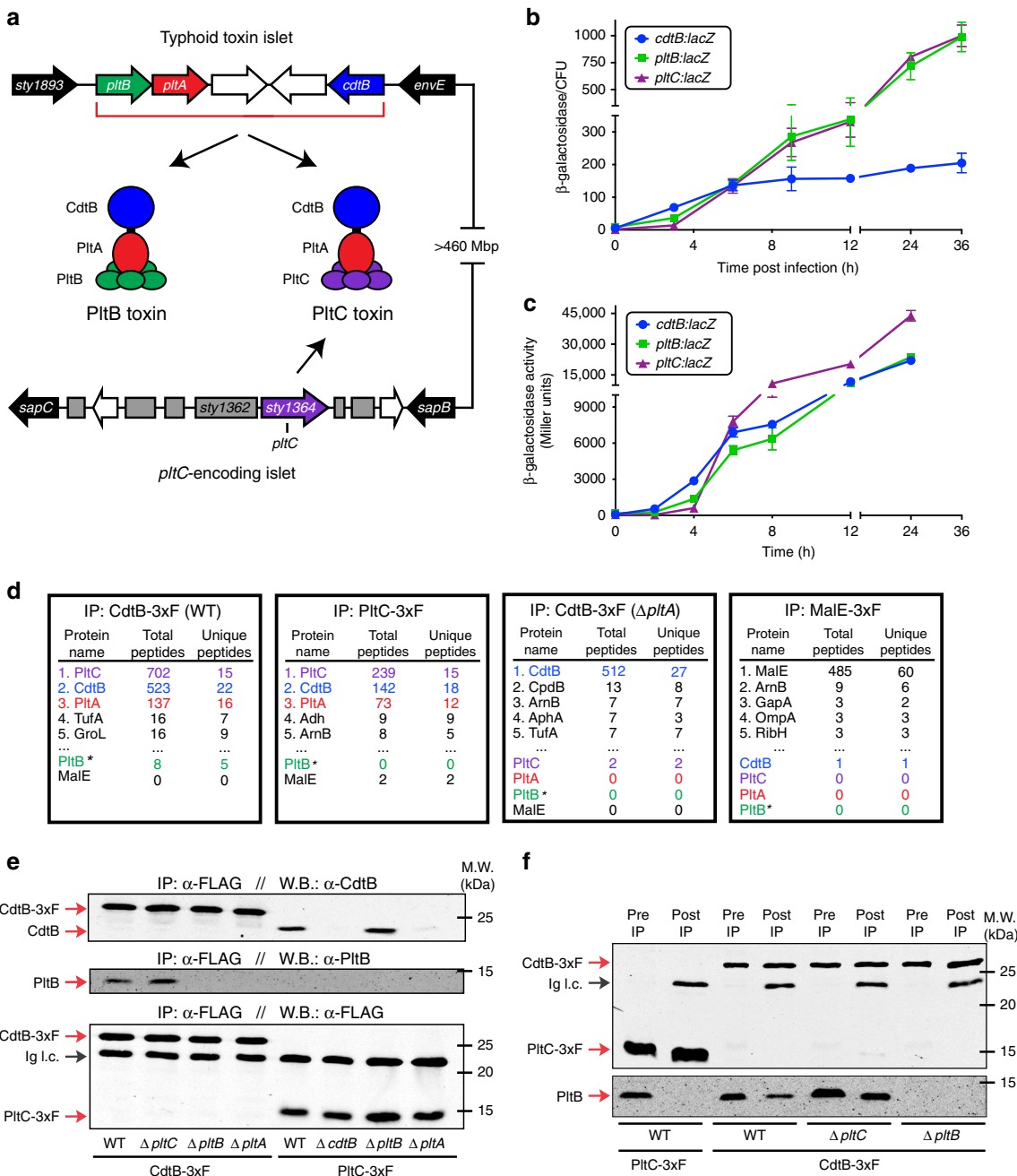

**Fig. 1** *S.* Typhi produces two distinct typhoid toxins with common active but different delivery subunits. **a** Illustration of the *S.* Typhi typhoid toxin genomic locus, as well as a distant locus that encodes *pltC* (*sty1364*), an orphan pertussis-like toxin delivery subunit that exhibits homology to *pltB*. **b**, and **c** Expression of *pltC*, *pltB* and *cdtB* over time under conditions that stimulate typhoid toxin gene expression. The β-galactosidase activity in the *pltC:lacZ*, *pltB:lacZ* and *cdtB:lacZ S.* Typhi reporter strains was measured at the indicated time points following infection of Henle-407 cells **b** or growth in TTIM medium **c**. Values indicate the mean ±S.D. for three independent samples. **d** Interaction of PltC with PltA/CdtB in *S.* Typhi grown in TTIM. Cell lysates from *S.* Typhi strains encoding *cdtB*-3xFLAG, *pltC*-3xFLAG, *cdtB*-3xFLAG (in Δ*pltA* background), or *malE*-3xFLAG were immunoprecipitated with an anti-FLAG antibody and interacting proteins were identified using LC/MS/MS. For each sample the number of peptides for the five most abundant proteins recovered and for all typhoid toxin subunits (color-coded according to panel **a**) are shown. *Detection of PltB with the LC–MS/MS protocol even for purified typhoid toxin preparations is inefficient. **e** *S.* Typhi produces both PltB- and PltC-typhoid toxins within infected human cells. Henle-407 cells were infected with *S.* Typhi wild type or the indicated mutant strains encoding 3xFLAG epitope-tagged CdtB or PltC (as indicated) and 24 hs post-infection the interaction of the indicated toxin components were probed by co-immunoprecipitation and western blot analysis. **f** PltB forms a complex with CdtB, but not with PltC. The interaction of the indicated toxin components in cell lysates of the indicated strains encoding 3xFLAG epitope tagged CdtB or PltC were probed by anti-FLAG co-immunoprecipitation and western blot analysis. Whole cell lysates (Pre IP) and immunoprecipitated samples (post IP) were probed using an anti-FLAG antibody as a control (top blot) and an anti-PltB antibody (bottom blot) to identify PltB interactions with CdtB or PltC in the indicated strains. Ig. l. c.: Immunoglobulin light chain detected by the secondary antibody. Source data are provided as a Source Data file

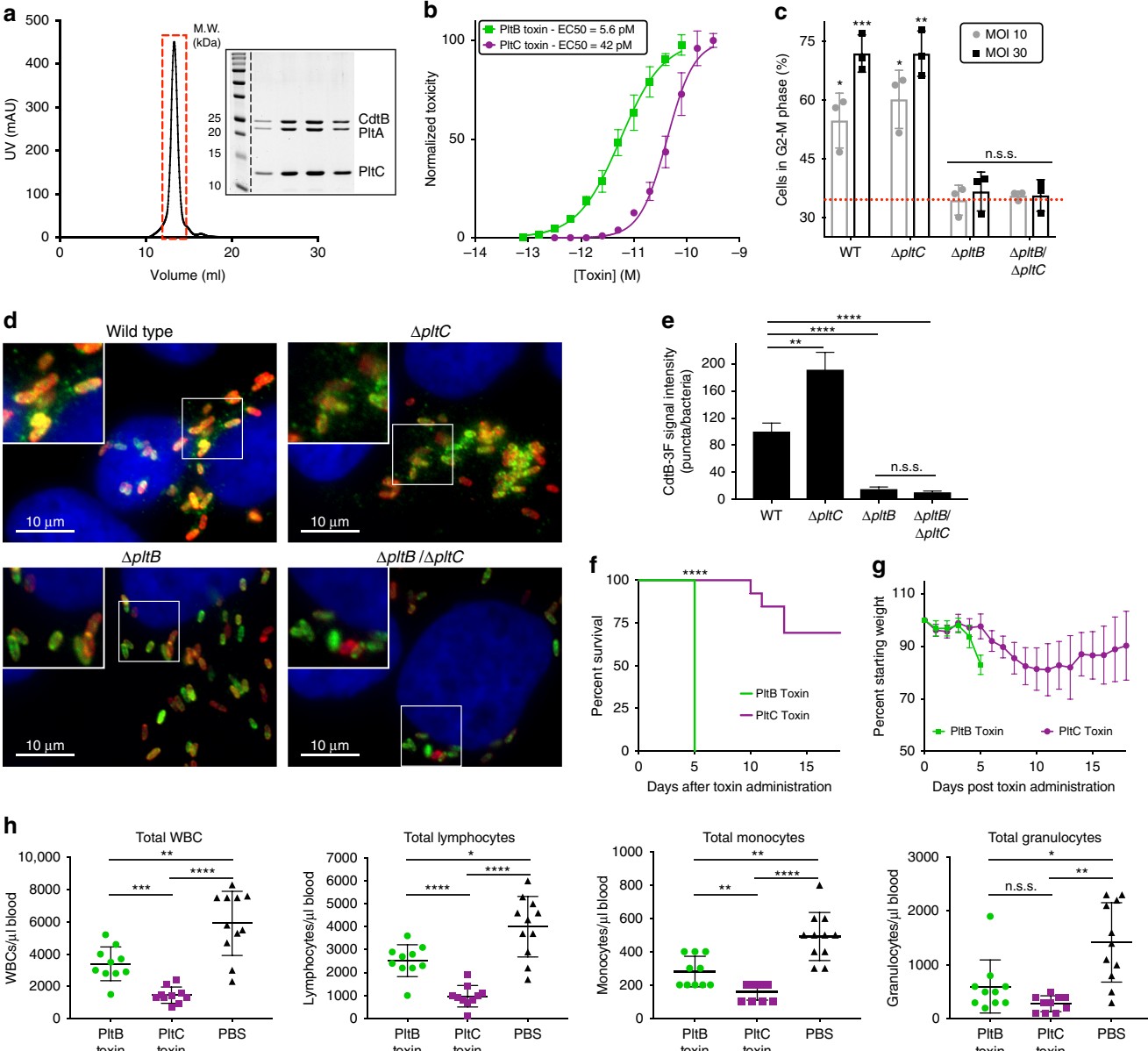

**Fig. 2** The PltB- and PltC-typhoid toxins exhibit different biological properties. **a** Gel filtration chromatography and SDS-PAGE/Coomassie blue (inset) analyses of purified PltC-typhoid toxin. Lanes on gel represent individual chromatographic fractions (red box) containing purified toxin. **b** PltC-typhoid toxin elicits G2/M cell cycle arrest in human epithelial cells. Purified PltB- or PltC-typhoid toxins were added to the culture medium of Henle-407 cells at the indicated concentrations and 48h after, cells were fixed and analyzed by flow cytometry to evaluate toxicity as indicated in Materials and Methods. The data shown are the mean normalized toxicity ± S.D. for three independent experiments. **c** PltC-typhoid toxin does not induce G2/M arrest in *S*. Typhi-infected cells. Henle-407 cells were infected with the indicated strains at a multiplicity of infection (MOI) of 10 or 30, as indicated, and 48h post-infection cells were collected and the percentage of cells in G2/M phase was determined as described for panel **b**. Mean values±S.D. are shown for three independent experiments assayed in duplicate (6 total samples). Asterisks denote statistically significant levels G2/M cell cycle arrest compared to mock infected cells (red dotted line) as determined by unpaired two-tailed *t*-tests. **d**, **e** PltC-typhoid toxin is not packaged into vesicle transport carriers. Henle-407 cells were infected with the indicated *S*. Typhi strains encoding 3x-FLAG epitope-tagged CdtB and 48 hs post-infection the cells were fixed and stained with DAPI (blue), α-FLAG (green), and α-*S*. Typhi LPS (red) antibodies. Typhoid toxin-containing export vesicles, which appear as green puncta **d**, were quantified by image analysis **e** as indicated in Materials and methods. Values are from >25 images (~100 infected cells) taken in two independent experiments and represent the mean relative ratios ± S.E.M. Asterisks denote the statistical significance of the indicated pairwise comparisons determined using unpaired two-tailed *t*-tests. **f**–**h** The PltB- and PltC-typhoid toxins elicit different effects when administered to mice. Highly purified preparations of PltB- (2μg) or PltC-typhoid toxins (10μg) were administered to C57BL/6 mice. For one group of mice, their survival **f** and body weight (mean ± S.D.) **g** was recorded at the indicated times. The remaining mice were killed at four days post-toxin administration and a blood sample was collected and analyzed to quantify the indicated cell types (mean ± S.D.) **h**. WBCs, white blood cells. The Mantel-Cox test was used for statistical analysis of mouse survival and Brown-Forsythe and Welch ANOVA coupled with Dunnett's T3 multiple comparisons tests were used to statistically compare the indicated samples for the blood analysis. For all panels, ****$p < 0.0001$, ***$p < 0.001$, **$p < 0.01$, *$p < 0.05$, n.s.s. not statistically significant. Source data are provided as a Source Data file

PltC we examined the ability of the purified toxin to intoxicate cultured cells as measured by cell cycle arrest at G2/M due to the DNA damage inflicted by the CdtB subunit (Fig. 2a, b)[5,11,15]. We found that PltC-typhoid toxin was able to intoxicate cultured epithelial cells in a similar fashion to the PltB version of the toxin although with a higher (~7 fold) EC50. Interestingly, however, cultured cells infected with a strain that lacks *pltC* (exclusively producing PltB-typhoid toxin) were intoxicated in a manner indistinguishable to cells infected with wild type *S.* Typhi, although cells infected with a Δ*pltB* mutant strain (exclusively producing PltC-typhoid toxin) did not exhibit detectable signs of intoxication (Fig. 2c). The observations that PltC-typhoid toxin is produced to significant levels during infection (Fig. 1e) and is able to intoxicate when directly applied to cultured cells, but it does not intoxicate during bacterial infection suggested that the two alternative forms of the toxin might differ in their delivery mechanisms after their synthesis by intracellular *S.* Typhi. We have previously shown that following its production, typhoid toxin is secreted from the bacterial periplasm into the lumen of the *Salmonella*-containing vacuole (SCV) by a specialized protein secretion mechanism involving a specialized muramidase that enables the toxin to cross to the trans side of the peptidoglycan (PG) layer, from where it can be released by various membrane-active agonists such as bile salts or anti-microbial peptides[16]. We found that both forms of typhoid toxin are released from the bacteria using this same mechanism (Supplementary Fig. 4). It has been shown that after its secretion into the lumen of the SCV, typhoid toxin is packaged into vesicle transport intermediates that carry the toxin to the extracellular space, a process that is orchestrated by interactions of its B subunit PltB with specific luminal receptors[5,17]. Therefore we examined whether differences in receptor specificity between PltB and PltC may lead to differences in their intracellular transport pathways after bacterial infection. We infected cultured cells with wild-type *S.* Typhi or isogenic mutants carrying deletions in *pltC*, *pltB*, or both, and monitored the formation of CdtB-containing transport carriers using an immunofluorescence assay (Fig. 2d, e). We found that toxin carriers were absent in cells infected with the Δ*pltB* strain although they were readily detected in cells infected with the Δ*pltC* mutant. These results indicate that the formation of the transport carriers is strictly dependent on PltB, presumably because the PltC version of typhoid toxin does not engage the sorting receptor. In fact, the level of transport carriers was measurably increased in cells infected with the Δ*pltC S.* Typhi mutant, an indication that the absence of PltC leads to the assembly of higher levels of export-competent PltB-containing typhoid toxin. Collectively, these data indicate that the two typhoid toxins differ significantly with respect to their ability to engage the sorting receptors within the lumen of the SCV leading to substantial differences in the export of the toxins to the extracellular space, a pre-requisite for intoxication after bacterial infection of cultured cells. These results also suggest that, during infection, the two toxins may exert their function in different environments and may target different cells.

To gain insight into potential differences between the activities of the two forms of typhoid toxin, we evaluated the consequences of systemically administering to C57BL/6 mice highly purified preparations of PltB- or PltC-typhoid toxins. We found that, although administering 2 μg of PltB-typhoid toxin was sufficient to kill all mice tested within five days, mice receiving 10 μg of PltC-Typhoid toxin (~5-fold more) survived for at least 10 days and the majority survived the full course of the experiment (Fig. 2f). Furthermore, in contrast to PltB-typhoid toxin[6], mice receiving the PltC-typhoid toxin showed no neurological symptoms, but did lose weight and showed signs of malaise and lethargy, although these symptoms were delayed and less

severe than those observed in the PltB-typhoid toxin treated animals (Fig. 2g). Peak toxicity for PltC-typhoid toxin treated mice was observed between 8 and 13 days post-administration, after which the majority of treated animals fully recovered. Interestingly, despite eliciting fewer and milder overt symptoms compared the PltB-typhoid toxin treated mice, PltC-typhoid toxin caused a significantly greater reduction in the numbers of total white blood cells, lymphocytes and monocytes (Fig. 2h). Collectively, these data indicate that the two typhoid toxins preferentially target different cells/tissues. Therefore producing two toxins variants confers functional versatility to typhoid toxin, presumably enabling *S.* Typhi to expand the spectrum of host cell targets that it can engage.

**The two typhoid toxins are differentially regulated**. Given the substantially different functional properties exhibited by the two forms of typhoid toxin, we reasoned that *S.* Typhi might have evolved regulatory mechanisms to preferentially produce the different forms of the toxin under different conditions. Expression of the typhoid toxin locus genes (i. e. *pltB*, *pltA* and *cdtB*) is controlled by the PhoP/PhoQ (PhoPQ) two-component regulatory system[12]. However, we found that in the absence of PhoPQ, *pltC* was robustly expressed during bacterial infection indicating that, despite exhibiting a similar intracellular expression pattern to the other typhoid toxin genes, the regulation of *pltC* expression must be distinct (Supplementary Figs. 5, 6). To decipher its regulatory network, we applied FAST-INseq[12] to screen for *S.* Typhi genes that influence *pltC* expression within infected cultured cells (Fig. 3a). Cultured epithelial cells were infected with a library of *S.* Typhi transposon mutants that encode a GFP reporter of *pltC* expression and, using fluorescence activated cell sorting (FACS), bacterial mutants that expressed *pltC* were separated from those that did not. Transposon insertion site sequencing (INseq)[18–20] was then used to identify transposon-disrupted genes that were over-represented in the bacterial population that did or did not express *pltC*, thus identifying candidate genes required for the regulation of *pltC* expression (Fig. 3b and Supplementary Table 1 and Supplementary Data 2). Notably, the most significantly enriched mutants that did not express *pltC* were insertions within *ssrA* and *ssrB*, which encode a two-component regulatory system (Supplementary Table 1 and Supplementary Data 2)[21–23]. This system is the master regulator of the expression of a type III protein secretion system encoded within *Salmonella* pathogenicity island 2, an essential *Salmonella* virulence factor that, like typhoid toxin, is selectively expressed by intracellular bacteria[24–26]. *S.* Typhi strains carrying deletion mutations in *ssrA/ssrB*, showed drastically reduced levels of *pltC* expression following infection of cultured epithelial cells, confirming the observations in the genetic screen (Fig. 3c, Supplementary Fig. 6). Interestingly, *phoP* and *phoQ* were also identified amongst the mutants that yielded reduced *pltC* expression. Examination of *pltC* expression at the population and single bacterium levels (Fig. 3c, Supplementary Figs. 5, 6) revealed that, although most Δ*phoPQ S.* Typhi express wild-type levels of *pltC* during infection, this mutant results in a larger population of intracellular *S.* Typhi that fail to express *pltC* (2.3% wild type vs. 8.8% Δ*phoPQ*, see Supplementary Fig. 6). Coupled with previous findings that PhoPQ can activate the expression of *ssrA/ssrB*, these data suggest that a small population of *S.* Typhi require PhoPQ activation in order to stimulate *ssrAB* expression under these infection conditions[26,27]. In stark contrast to what we observed for *pltC*, the absence of SsrA/SsrB had a negligible effect on *pltB* expression (Fig. 3c, Supplementary Figs. 5, 6). Collectively, these results suggest that the SsrA/SsrB two-component system is the principal activator of *pltC*

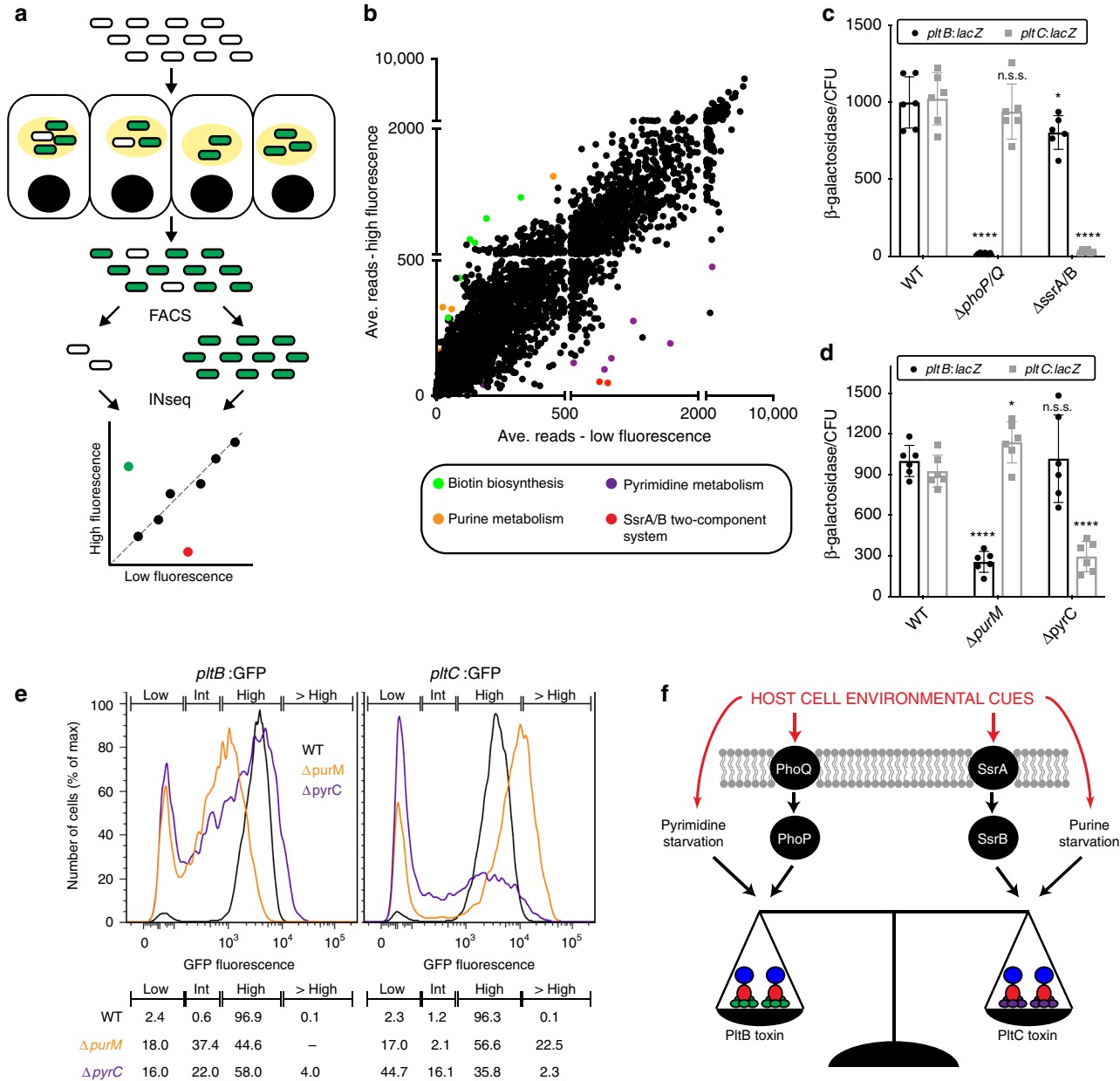

**Fig. 3** Distinct regulatory mechanisms and metabolic cues control the expression of PltB and PltC. **a** Schematic representation of the FAST-INseq genetic screen used to identify *S.* Typhi genes that influence *pltC* expression in infected host cells. A large library of random transposon mutants was generated in the *S.* Typhi *pltC:gfp* strain and used to infect Henle-407 cells. Sixteen hours post-infection the bacteria were isolated and sorted by FACS into pools exhibiting high and low GFP fluorescence. INseq was then used to identify mutants that stimulate (green dot in plot) or reduce (red dot) *pltC* expression during infection. **b** Overview of the results of the FAST-INseq screen. Plot shows the normalized numbers of sequencing reads of transposon insertions within each *S.* Typhi gene in the high fluorescence vs. low fluorescence pools. **c**, **d** Expression levels of *pltB:lacZ* and *pltC:lacZ* reporters in infected human cells for wild-type *S.* Typhi (WT) and the indicated deletion mutant strains. Henle-407 cells were infected with the indicated strains for 24 h, after which the β-galactosidase activity from bacterial lysates was measured and normalized to the numbers of CFU recovered. Values indicate mean values ± S.D. for six independent determinations taken over two separate experiments. Asterisks denote statistically significant differences relative to the corresponding wild-type sample determined using unpaired two-tailed *t*-tests. \*\*\*\**p* < 0.0001, \**p* < 0.05, n.s.s. not statistically significant. **e** Flow cytometry analysis of *pltB:gfp* and *pltC:gfp* expression of the indicated *S.* Typhi strains 24 h post-infection. Histograms show the GFP fluorescence intensities of individual bacteria for the indicated strains. Gates were established to show the percentage of bacteria exhibiting high, low and intermediate (int) fluorescence. The percentage of bacteria with fluorescence intensities within these gates is shown (bottom). Gating strategy provided in Supplementary Fig. 5b. **f** Overview of the identified factors that differentially affect the expression of *pltB* and *pltC* and thus are likely to be important for controlling relative abundance of the two typhoid toxins produced by *S.* Typhi upon encountering different environments during infection

expression during infection and indicate that the production of the two typhoid toxin delivery subunits is controlled by different, intracellularly-induced, global regulatory networks.

Our screen also identified several mutants that led to increased *pltC* expression. Among these mutants were insertions within all of the genes required for the biosynthesis of the cofactor biotin (Fig. 3b and Supplementary Table 2 and Supplementary Data 2), which based on previous studies[12], are likely to affect expression of *pltC* indirectly, by preventing the expansion of a population of cytosolic bacteria (i. e. located outside of the *Salmonella* containing vacuole) unable to express typhoid toxin. These results therefore suggest that, like the other typhoid toxin genes, *pltC* expression also requires the specific environment of the *Salmonella* containing vacuole. We also found that insertions within genes required for purine biosynthesis resulted in increased *pltC* expression, while insertions within pyrimidine biosynthesis genes had the opposite effect (Fig. 3b, Supplementary Tables 1 and 2 and Supplementary Data 2). This is particularly noteworthy since purine biosynthesis mutants result in reduced expression of *pltB*[12]. Follow up experiments exploring the expression of *pltB* and *pltC* in cultured cells infected with isogenic purine *(ΔpurM)* or pyrimidine *(ΔpyrC)* biosynthesis mutants confirmed the results of our screen and demonstrated that, in intracellular *S.* Typhi, purine limitation favors *pltC* expression while pyrimidine limitation favors *pltB* expression (Fig. 3d, e). Further experiments will be required to decipher the nature of this regulation, however a wide range of regulatory mechanisms have been described involving nucleobases/nucleotides and second messengers derived from these molecules, many of which operate post transcriptional initiation[28–31]. Given that levels of purines and pyrimidines are connected through their common use in DNA/RNA synthesis, it is tempting to speculate that the inverse effects observed for *pltB* and *pltC* expression may be due to a single regulatory molecule that exerts opposite effects on the expression of the two delivery subunits. Collectively, these results demonstrate that the relative expression levels of the two typhoid toxin delivery subunits are substantially altered in response to low nucleotide concentrations, and suggest that different nutrient availability may serve as a cue that enables *S.* Typhi to adjust the balance of the two forms of typhoid toxin it produces within a given environment (Fig. 3f).

## Discussion

We have shown here that *S.* Typhi produces two different versions of typhoid toxin that share their enzymatic subunits but utilize alternative delivery subunits resulting in substantially different biological activities. In particular, our results indicate that PltB-typhoid toxin is more efficient at causing neurological symptoms, which are associated with increased lethality. In contrast, PltC-Typhoid toxin is more effective at targeting white blood cells as mice challenged with this form of typhoid toxin exhibited a more pronounced leukopenia. Therefore, by assembling toxins with different targeting mechanisms, *Salmonella* Typhi may be able to target a broader array of cell types in different tissue environments.

The typhoid toxin locus has a sporadic distribution in the *Salmonella* genus and is found in a range of different genome locations, often within prophage. Given these factors, it is noteworthy that virtually all typhoid toxin-encoding strains whose genomes have been sequenced also encode a second B subunit homolog (Supplementary Fig. 1)[9,10,32]. This strongly suggests that producing multiple forms of typhoid toxin is not unique to the Typhi serovar, but rather is an integral aspect of typhoid toxin biology; indeed, genetic evidence suggests that an orthologous B subunit is also important for the function of the typhoid toxin

produced by the Javiana serovar[32]. The distributions and genomic locations of these elements indicate that the acquisition of the core typhoid toxin islet and the second B subunit occurred in separate evolutionary events in several distinct *Salmonella* lineages, implying that producing functionally diverse typhoid toxins imparts a strong evolutionary advantage to ecologically diverse salmonellae. More broadly, we have also identified other pathogens that encode orphan B subunits that are homologous to components of an $AB_5$ toxin located elsewhere in their genomes, suggesting that the assembly of alternate toxins may be a more general feature of $AB_5$ toxins (Supplementary Fig. 7). For example, in addition to the locus encoding pertussis toxin, *Bordetella pertussis* encodes two orphan orthologs of its B subunits at a distant genome location. Under certain conditions, these proteins have been reported to be co-synthesized and co-secreted with other pertussis toxin components[33].

Here, we show that B subunits with significantly different amino acid sequences encoded by the same bacterium assemble with common A subunits into distinct toxins with different biological properties. This "lego-like" assembly of structurally-similar but functionally distinct components, which genomic evidence suggests is likely to be conserved in diverse bacterial lineages encoding different $AB_5$ toxins, suggests that the evolutionary expansion of the $AB_5$ class of toxins was likely fueled by the plasticity inherent to their structural design coupled with the functional versatility that can be achieved through combining homologous toxin components.

## Methods

**Bacterial strains and cell lines**. *S.* Typhi strains employed in this study were derived from the wild-type isolate ISP2825[34] and were constructed using standard recombinant DNA and allelic exchange procedures using the *E. coli* β-2163 *Δnic35* as the conjugative donor strain[35]. All the *S.* Typhi deletion mutant strains carry deletions of the entire coding regions of the indicated gene or genes from the start to the stop codons. The *malE*-3xFLAG strain was generated by moving a C-terminal 3xFLAG tagged version of *malE* to the *pltC* locus (resulting in a *pltC* deletion). All other strains featuring 3xFLAG epitope-tagged genes were generated by replacing the native gene with a C-terminal 3xFLAG tagged version at its native genomics locus. A complete list of bacterial strains used in this study is provided in Supplementary Table 3. Strains were routinely cultured in LB broth at 37 °C. For *in vitro* growth assays that employed typhoid toxin inducing growth medium (TTIM) a previously described chemically defined growth medium was used[12] that was based on N minimal medium[36]. All experiments using cultured cells were conducted using the Henle-407 human epithelial cell line, which was obtained from the Roy Curtiss library collection. Cells were cultured in Dulbecco's modified Eagle medium (DMEM, GIBCO) supplemented with 10% Fetal Bovine Serum (FBS) at 37 °C with 5% $CO_2$ in a humidified incubator. This cell line was routinely tested for mycoplasma contamination using a Mycoplasma Detection Kit (SouthernBiotech, Cat# 13100–01).

***Salmonella* Typhi infections**. To infect Henle-407 cells, overnight cultures of *S.* Typhi were diluted 1/20 into fresh LB containing 0.3 M NaCl and grown to an $OD_{600}$ of 0.9. Cells were infected for 1 h in Hank's balanced salt solution (HBSS) at the indicated multiplicity of infection (MOI). Cells were then washed three times with HBSS and incubated in culture medium containing 100 μg/ml gentamycin to kill extracellular bacteria. After 1 h, cells were washed and fresh medium was added containing 5 μg/ml gentamycin to avoid repeated cycles of reinfection.

**β-galactosidase assays**. For *in vitro* grown samples, overnight cultures were washed two times with TTIM, diluted 1/20 into fresh TTIM and grown for the indicated amount of time at 37 °C at which point 10 or 20 μl of the culture was added to 90 μl of permeabilization buffer (100 mM $Na_2HPO_4$, 20 mM KCl, 2 mM $MgSO_4$, 0.8 mg/ml hexadecyltrimethylammonium bromide [CTAB], 0.4 mg/ml sodium deoxycholate, 5.4 μL/ml β-mercaptoethanol) and assayed, as described below. For samples collected from infected cells, $3 \times 10^5$ Henle-407 cells were plated in 6-well plates and grown for 24 h prior to infection with the indicated strains. Following infections, cells were washed two times with PBS, removed from the plates using dilute trypsin and pelleted by centrifugation for 5 min at $150 \times g$. Cells were then lysed in 0.1% sodium deoxycholate (in PBS) supplemented with 100 μg/ml DNase I and the lysate was centrifuged at $5000 \times g$ for 5 min to pellet the bacteria. The bacteria were re-suspended in PBS, a small aliquot of which was diluted to calculate the total number CFU recovered. The remainder of the bacteria were pelleted, re-suspended in 100 μl of permeabilization buffer and assayed as

described below. Assays were conducted at 24 h post infection (hpi) unless otherwise indicated. β-galactosidase assays were conducted using a modified version of the protocol developed by Miller[12,37]. Briefly, samples were permeabilized for 20 min at room temperature in the buffer described above after which 600 μl of substrate buffer (60 mM $Na_2HPO_4$, 40 mM $NaH_2PO_4$, 2.7 μL/mL β-mercaptoethanol, 1 mg/ml ONPG [o-nitrophenyl-β-D-galactopyranoside, Sigma]) was then added to initiate the reactions. Once the samples developed an obvious yellow color, the reactions were quenched using 700 μl of 1 M $Na_2CO_3$ and the reaction time was noted. Cell debris was removed by centrifugation at $20,000 \times g$ for 5 min and the $OD_{420}$ of the samples was measured. Miller units were calculated as: $(1000 \times OD_{420})/($reaction time [minutes] × culture volume assayed [ml] × $OD_{600}$ [culture]).

**Co-immunoprecipitation experiments**. To identify interaction partners for the various typhoid toxin subunits, S. Typhi bacterial cell lysates were immunoprecipitated using C-terminal 3xFLAG epitope-tagged CdtB or PltC (tags were incorporated at the native genomic locus) as indicated and the eluates were analyzed by LC–MS/MS or western blot. A strain encoding a C-terminal 3xFLAG epitope-tagged version of MalE, a periplasmic protein that is expressed in TTIM, which was cloned in place of pltC at the pltC locus in the S. Typhi chromosome and included as a negative control for the LC–MS/MS analysis. For in vitro grown samples, the indicated strains were grown overnight in LB, washed twice using TTIM, diluted 1/20 into 12 ml of fresh TTIM and grown overnight. Cultures were pelleted, re-suspended in lysis buffer (50 mM Tris pH 7.5, 170 mM NaCl, cOmplete mini protease inhibitors [Sigma], 40 ug/ml DNase I) and lysed using the One Shot cell disruption system (Constant Systems, Ltd). Clarified lysates were immunoprecipitated overnight at 4 °C using ANTI-FLAG M2 affinity gel (Sigma). Immunoprecipitated samples were washed thoroughly using 50 mM Tris pH 7.5/170 mM NaCl/50 mM galactose/0.1% Triton X-100 and eluted using 0.1 M glycine-HCl (LC–MS/MS) or SDS–PAGE loading buffer (western blot analysis). For LC–MS/MS analysis, the eluted samples were precipitated overnight at −20 °C in 80% acetone and washed twice using 80% acetone. The samples were then reduced using DTT, alkylated using iodoacetamide and trypsin digested overnight. C18 column purified peptides were then analyzed by LC/MS/MS, as previously described[38] and MS/MS scans were processed and searched using MASCOT (Matrix Science Ltd.). The resulting peptide and protein assignments were filtered to keep only those identifications with scores above extensive homology. For western blot analysis, eluted samples were run on 10–15% SDS–PAGE and transferred to nitrocellulose membranes. Membranes were blocked using 5% non-fat milk, incubated overnight at 4 °C with the indicated primary antibody followed by a 2 h incubation with a fluorescently conjugated α-mouse (FLAG) or α-rabbit (PltB) secondary antibody and analyzed using the Li-Cor Odyssey blot imager. The following antibodies were used for western blot analysis: α-FLAG mouse monoclonal (Sigma, F3165, 1:5000 dilution), α-PltB rabbit polyclonal (produced for Galan lab by Pocono Rabbit Farm & Laboratory, Inc. Canadensis PA, 1:5000 dilution), α-CdtB rabbit polyclonal (produced for Galan lab by Pocono Rabbit Farm & Laboratory, 1:5000 dilution). Uncropped gel images, as well as unedited and uncropped gel images, for these (and all other relevant) experiments are provided in the Source Data file.

For samples isolated from infected cells, three 15 cm dishes containing a total of $7.5 \times 10^7$ Henle-407 cells were infected at an MOI of 20 as described above using each of the indicated S. Typhi strains. 24 h post-infection, cells were washed twice using PBS and collected using a cell scraper, pelleted and washed once using PBS. Cell pellets were then resuspended in lysis buffer (50 mM Tris pH 7.5, 170 mM NaCl, complete mini protease inhibitors [Sigma], 40 ug/ml DNase I, 10 mM N-ethylmaleimide) and lysed using a Branson Digital Sonifier (3 s on/8 s off, 35% amplitude, 3 min total). Clarified lysates were then immunoprecipitated and analyzed by western blot as described above.

**Typhoid toxin purification**. Both typhoid toxins were purified according to a previously established protocol[6]. Briefly, pltB, pltA and cdtB-6xHis (PltB-typhoid toxin) or pltC, pltA and cdtB-6xHis (PltC-typhoid toxin) were cloned into pET28a(+) vector (Novagen). E. coli strains carrying these expression vectors were grown to $OD_{600}$ ~0.8, at which time 250 μM IPTG was added to induce the expression of the toxin genes and the cultures were grown overnight at 30 °C. Bacterial cells were pelleted by centrifugation and lysed. Crude lysates were affinity purified using Nickel resin (Qiagen), followed by cation exchange chromatography using a Mono S column (Sigma-Aldrich) and finally gel filtration using a Superdex-200 column (Sigma-Aldrich). The final fractions were analyzed by SDS–PAGE to confirm purity.

**Culture cell intoxication assays**. To assess typhoid toxin toxicity in cultured human cells, the number of cells arrested in G2/M (as a consequence of CdtB-mediated DNA damage) was determined using flow cytometry as previously described[6]. For experiments using purified toxin, $2.5 \times 10^4$ Henle-407 epithelial cells were plated in 12-well plates. After 24 h the cells were washed and fresh medium containing the indicated concentrations of PltB- or PltC-typhoid toxin were added. Forty eight hours later, the cells were removed from the dishes using trypsin treatment, pelleted, re-suspended in 300 μl PBS, fixed by adding ice cold

ethanol dropwise to a final concentration of 70% and incubated overnight at −20 °C. Cells were then washed with PBS, re-suspended in 500 μl of PBS containing 50 μg/ml propidium iodide, 0.1 μg/ml RNase A and 0.05% Triton X-100 and incubated for 30 min at 37 °C. Cells were then washed and re-suspend in PBS, filtered and analyzed by a flow cytometry. The DNA content of cells was determined using FlowJo (Treestar). In order to obtain $EC_{50}$ values, the percentage of cells in the G2/M phase (% G2/M) was determined for each sample and converted to normalized toxicity by subtracting the % G2-M value observed of untreated cells and dividing by (maximum % G2/M - untreated % G2/M). The maximum % G2/M value was considered to be 90% based on our experience that values above this number are not reliably observed even at saturating toxin concentrations. For experiments using S. Typhi infected cells, $2.5 \times 10^4$ Henle-407 cells were plated in 12-well plates and were infected using the indicated S. Typhi strains at the indicated MOI values 24 h later. After 48 h post-infection the cells were collected, fixed, stained and analyzed as described above.

**Typhoid toxin secretion assay**. To assess the mechanism of secretion for the PltB- and PltC typhoid toxins, an in vitro toxin secretion assay was employed to determine whether toxin secretion from $\Delta pltB$ (producing exclusively PltC-typhoid toxin) and $\Delta pltC$ (producing exclusively PltB-typhoid toxin) S. Typhi strains was dependent upon both TtsA and outer membrane perturbation, in accordance with the recently described mechanism of typhoid toxin secretion[16]. The indicated strains, all of which encode 3x-FLAG epitope-tagged cdtB from its native genomic locus, were grown overnight in LB, washed twice using TTIM, diluted 1/20 into fresh TTIM and grown for 24 h at 37 °C to induce typhoid toxin and ttsA expression. The bacteria were then pelleted, washed thoroughly, and incubated for 15 min either in 0.075% bile salts (Sigma) in PBS or in PBS alone (mock treated). The bacteria were then pelleted and the supernatants were filtered using a 0.2 μm filter and TCA precipitated. The amount of toxin in the pellet and supernatant fractions was then determined by western blot analysis using an M2 α-FLAG antibody as described above.

**Immunofluorescence microscopy assay for typhoid toxin export**. To compare the export of typhoid toxin from the Salmonella-containing vacuole to the extracellular space for PltB-typhoid toxin and PltC-typhoid toxin, we employed a previously established immunofluorescence-based assay that quantifies the levels typhoid toxin that are within exocytic vesicles compared to the levels associated with bacteria[17]. Henle-407 cells plated on glass coverslips were infected with the indicated cdtB-3xFLAG epitope-tagged S. Typhi strains at an MOI of 10. Forty eight hours post-infection, the samples were fixed in 4% paraformaldehyde and blocked using 3% BSA/0.3% triton X-100 in PBS. The coverslips were then incubated with a 1:5000 dilution of mouse monoclonal M2 anti-FLAG antibody (Sigma) and a 1:10,000 dilution of rabbit polyclonal anti-S. Typhi LPS antibody (Sifin) overnight at 4 °C. After thoroughly washing in PBS, samples were then stained using Alexa 488-conjugated anti-mouse and Alexa 594-conjugaed anti-rabbit antibodies and DAPI (Sigma) for 2 h at room temperature, washed extensively using PBS, mounted on coverslips, and imaged imaged using an Eclipse TE2000 inverted microscope (Nikon) with an Andor Zyla 5.5 sCMOS camera driven by Micromanager software (https://www.micro-manager.org).

The open source software ImageJ (http://rsbweb.nih.gov/ij/) was used to quantify toxin export in images captured in random fields as described previously[17]. Briefly, the LPS signal was used to identify the bacterial cells and the CdtB-3xFLAG signal was used to identify typhoid toxin. The typhoid toxin signal within the area associated with bacterial cells was subtracted from the total typhoid toxin signal in order to obtain the signal associated with typhoid toxin carrier intermediates. In a given field, the fluorescence intensity of the typhoid toxin signal that was associated with toxin carriers was normalized to the bacterial-associated typhoid toxin signal within the same field.

**Animal intoxication experiments**. All animal experiments followed the ethical regulations and were conducted according to protocols approved by Yale University's Institutional Animal Care and Use Committee. Prior to beginning this study, approval to conduct the animal experiments presented was granted by this committee. C57BL/6 mice were anesthetized with 30% w/v isoflurane in propylene glycol and 100 μl of toxin solution containing the indicated concentration of purified PltB- or PltC-typhoid toxin was administered via the retro-orbital route. Changes in behavior, weight and survival of the toxin-injected mice were closely monitored for the duration of the experiment. Blood samples were collected by cardiac puncture 4 days after toxin administration in Microtainer tubes coated with EDTA, kept at room temperature and analyzed within 2 h after blood collection using a HESKA Veterinary Hematology System.

**FAST-INSeq screen**. The FAST-INSeq screen was employed to identify S. Typhi genes that, when disrupted by transposon mutagenesis, lead to altered expression of a pltC:gfp reporter within infected Henle-407 epithelial cells. This screen was conducted as previously described[12] using a pltB:gfp reporter. Transposon mutagenesis was conducted using a mariner transposon delivered by pSB4807, which was mobilized into the pltC:gfp S. Typhi strain by conjugation using E. coli β-2163 Δnic35 as the donor strain. Mutants were selected by plating on LB agar plates

containing 30 µg/ml chloramphenicol. A total of ~150,000 mutants were collected and multiple aliquots of this library were stocked for subsequent use in the screen. For each iteration of the screen, five 15 cm dishes were seeded with $1 \times 10^7$ Henle-407 cells each and grown for ~24 h prior to infection using the transposon mutant library described above. Aliquots of the inoculum used for infection were collected for subsequent INSeq preparation (inoculum pool). 18 h post infection, the cells were washed three times with PBS, detached from plates using dilute trypsin and pelleted by centrifugation at $150 \times g$ for 5 min. Cells were lysed using a 5 min incubation in 0.1% sodium deoxycholate (in PBS) supplemented with 100 µg/ml DNase I to degrade and solubilize the genomic DNA released from lysed host cells. The lysate was centrifuged at $5000 \times g$ for 5 min to isolate S. Typhi from the soluble cellular debris. The S. Typhi-containing pellet was re-suspended in PBS and further purified from cellular debris using two spins at $150 \times g$ for 5 min (discarding the pellet fraction) and one spin at $5000 \times g$ for 5 min (discarding the supernatant). An aliquot of the recovered S. Typhi was amplified by growth in LB at 37 °C and subsequently prepared for INSeq sequencing (post-infection pool) and the remainder was washed and diluted in PBS to a concentration of ~$2 \times 10^6$ bacteria/ml for FACS. A total of ~$2 \times 10^7$ S. Typhi mutants were sorted according to their fluorescence intensity in the GFP channel (488 nm excitation, 515/20 with 505LP emission) using a BD FACS Aria II flow cytometer. The isolated low fluorescence and high fluorescence pools were amplified by growth in LB at 37 °C and subsequently prepared for INSeq sequencing. The screen was conducted using the same mutant library on two independent occasions, and the low fluorescence pools from these two sorts were pooled and re-sorted (re-sort of low fluorescence populations).

For INSeq sequencing, genomic DNA was extracted from each of the mutant pools, digested with MmeI (New England Biolabs) and barcoded samples were prepared for sequencing as described previously[20]. The purified 121 bp DNA products containing barcodes to identify the individual mutant pools were sequenced on an Illumina HiSeq2000 system at the Yale Center for Genomic Analysis. The sequencing data were analyzed using the INSeq_pipeline_v3 package[19], which separated sequencing reads by pool, mapped/quantified insertions and grouped insertions by gene. For each pool, the total number of sequencing reads was normalized to be 2,176,000 (an average of 500/gene). To identify genes in which insertions were enriched in a statistically significant manner in one pool compared to another, a value of 50 (10% of the average number of reads per gene) was added to the normalized number of reads in both pools. Ratios of the log-transformed read numbers for the two pools were then calculated. Genes with values that were an average of more than two standard deviations from the mean over the two primary replicates of the screen and more than one standard deviation from the mean in all three sorts were considered to be significantly enriched.

**Analytical flow cytometry.** To probe pltB and pltC expression within infected cells at the single bacterium level, we employed a previously developed flow cytometry-based assay[12]. The indicated pltC:gfp and pltB:gfp strains carrying a plasmid driving constitutive mCherry expression were used to infect Henle-407 cells as described above. At 24 h post infection, bacteria were isolated and prepared for flow cytometry as described above for the FAST-INSeq screen. At this time point, which was chosen to capture a state of purine/pyrimidine starvation in the purM/pyrC mutants, we find very few instances of S. Typhi within the host cell cytosol and thus virtually all S. Typhi express high levels of both pltB and pltC in the wild-type strains[12]. For each sample the fluorescence intensity in the GFP channel (488 nm excitation, 515/20 with 505LP emission) was analyzed for at least 5000 mCherry-positive particles (532 nm excitation, 610/20 with 600LP emission) using a BD FACS Aria II flow cytometer. All samples were prepared and analyzed by flow cytometry in parallel. High and low fluorescence populations were defined based on the peaks observed in the wild-type samples for the given reporter strain. The intermediate population was defined as having a fluorescence intensity between the low and high gates and the "greater than high fluorescence" population was defined as having a fluorescence intensity greater than the high fluorescence peak ( >99.9% of particles in the wild-type sample). A figure exemplifying the gating strategy is provided in the Supplementary Information (Supplementary Fig. 5b).

**Unique biological materials.** All unique biological materials used in this study are available from the authors.

**Reporting Summary.** Further information on research design is available in the Nature Research Reporting Summary linked to this article.

## Data availability
The data that support the findings of this study are available within the paper, including its Supplementary Information file. The source data underlying Figs. 1b, c, e, f, 2a, b, e, g and Supplementary Figs. 2 and 4 are provided as a Source Data file.

## Code availability
Analysis of the raw FAST-INSeq sequencing data was facilitated by the INSeq_pipeline_v2 software package, which is available from the Andrew Goodman laboratory, Yale University.

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

## Acknowledgements

We thank members of the Galán laboratory for careful review of this manuscript. C.C.F. was supported in part by a Postdoctoral Fellowship from the Canadian Institutes of Health Research (CIHR). This work was supported by National Institute of Allergy and Infectious Diseases grant AI079022 (to J.E.G.).

## Author contributions

C.C.F. performed the majority of the experiments, analyzed data and was involved in project conception and design; G.S. carried out the animal experiments; M.L.T. performed the LC–MS/MS analyses; X. J. provided purified toxin preparations; J.E.G. analyzed data and was involved in project conception and design. C.C.F. and J.E.G. wrote the paper with comments from the authors.

## Additional information

**Competing interests:** The authors declare no competing interests.

