## [Peer Review File · Nature Communications]

Reviewers' comments:

Reviewer #1 (Remarks to the Author):

This is an intriguing paper that describes the identification of a second delivery subunit for typhoid toxin. The finding of a second B delivery system for a toxin is novel. Of further interest was the apparent differential *in vivo* regulation of the two delivery systems. The overall evidence for the alternate B subunit is persuasive, and evidence is provided that the toxin with PltC as a delivery subunit caused morbidity and some mortality when administered in purified form. However, in Figure 2, part h, the unpaired, two-tailed t test is not appropriate for comparing pair-wise when there are three groups. (A statistician may need to be consulted because ANOVA may also not be the correct test due to the variability in the PBS control sample.) Therefore, the concern is that the role of the PltC-delivered toxin in the alteration of blood cell counts may not be significant once the data are re-analyzed. In addition, a few other items should be addressed, and are detailed below.

1. There is no evidence in Figure 1d for the lack of heteropentamers since no PltB is found for any of the samples in that figure. Therefore, on page 4 line 6 should say (Fig 1e,f) as the support for the lack of heteropentamers.
2. A further explanation or speculation for the inability to detect PltB following immunoprecipitation and LC/MS/MS as shown in Figure 1d is warranted (in the text and not the figure legend) because PltB is only 4 aa shorter than PltC and is 47% similar according to the supplemental data, and PltC was detected easily.
3. It appears that the A2 moiety of the toxin may have higher affinity for PltC - based on finding more PltC in Fig1d and because when pltC is mutated, more PltB is detected at least in TTIM conditions. Therefore, the question of heterodimers may not be settled until PltB is immunoprecipitated and then PltC is checked for by western blot. However it does appear that the majority of the toxins are in homopentamers.
4. There appears to be more PltB immunoprecipitated in the absence of PltC when the cells were grown in the TTIM (Fig 1f) but not the Henle-407 cells (Fig 1e) - or at least much less of a difference, so the statement on p4, line 8 should indicate that the difference is when cells are grown in TTIM, or else the differences should be quantified.

5. There appears to be more PltB in the pltC mutant in Figure 1f in both pre and post conditions, although the CdtB-3xF levels appear to be similar. Is this difference meaningful? Does the pltC mutant retain the promoter? The methods section does not describe the nature of the mutations. This information could be added to the extended data.

6. Figure 1f needs more description and labeling. What do the "Pre" and "Post" labels mean (pre- and post-immunoprecipitation, presumably)? What was the top panel probed with, anti-Flag?

7. p6, second paragraph, line 6 states that pltC is expressed in cells in the absence of PhoPQ. Extended Table 1 shows phoP and phoQ as the 3rd and 4th genes "important for pltC expression in infected human cells". So, does that mean that the cut-off for "important" for expression should be above a ratio of 6?

Minor comments

1. Fig 1d- for the top label in each box, it should be indicated that each of those proteins has a 3xF tag.

2. page 24 line 5 - "dividing" rather than diving.

3. page 25 MOI rather than "moiety of infection"

Reviewer #2 (Remarks to the Author):

The author demonstrate that an "Orphan" B-subunit gene (PltC) in fact assembles into an active toxin that displays a different expression pattern and host cell range from PltB. This is very novel discovery and will be of interest to Salmonella researchers and others.

The work is convincing, with one minor exception. On page 4, while the trend is there, if you want to say PltB and PltC compete for assembly with the A-subunit you would have to do quantitative

western blots with multiple independent repeats. If they do not wish to do these experiments, they could just reword this section.

Page 14. Please clarify, "Mean values +/- S.D. are shown for three independent experiments assayed in duplicate (6 total samples)." Did you use N=6 (not correct), or average the duplicates and use N=3 (correct)?

Regarding Figure 3f, pyrimidines and purines seem like two sides of the same coin, I would appreciate some speculation as to why they act as different regulatory signals (purines are also needed for energy?)

Minor issues for clarity:

1. The text on page 3 uses "sty1364", but on figure 1 it is labeled PltC. On page 4, the big reveal is they are the same. To prevent reader confusion, tentatively call it PltC from the beginning.
2. Figure 2a insert. What are the lanes?
3. Figure 1a, extended data figure 5, pltC looks blue not purple.
4. Page 5, please define PG.

Response to the Reviewers' comments:

Reviewer #1:

This is an intriguing paper that describes the identification of a second delivery subunit for typhoid toxin. The finding of a second B delivery system for a toxin is novel. Of further interest was the apparent differential in vivo regulation of the two delivery systems. The overall evidence for the alternate B subunit is persuasive, and evidence is provided that the toxin with PltC as a delivery subunit caused morbidity and some mortality when administered in purified form. However, in Figure 2, part h, the unpaired, two-tailed t test is not appropriate for comparing pair-wise when there are three groups. (A statistician may need to be consulted because ANOVA may also not be the correct test due to the variability in the PBS control sample.) Therefore, the concern is that the role of the PltC-delivered toxin in the alteration of blood cell counts may not be significant once the data are re-analyzed. In addition, a few other items should be addressed, and are detailed below.

We thank the reviewer for this comment. Indeed, the reviewer is correct and the use of the use of multiple *t* tests is not appropriate in this context. We have looked into this matter and have used ANOVA to analyze these data. However, as the reviewer correctly points out, the variances are not equal due to the spread in the PBS control, so rather than using a standard ANOVA/post hoc test (such as Tukey) to determine significance, we have used Brown-Forsythe and Welch ANOVA tests coupled with Dunnett's T3 multiple comparisons tests, which do not assume equal variances in all samples. We are now confident that we have used an appropriate statistical test to analyze these data and have adjusted the *p* value thresholds (i.e. **, ***, etc) in the figure where appropriate. Importantly, the new analyses had a very modest effect on the *p* values. None of the values that met our significance cut off initially failed to do so using the new analysis and, in fact, none of the *p* value thresholds comparing the PltB and PltC toxins (the most relevant here) were affected after the new analyses.

1. There is no evidence in Figure 1d for the lack of heteropentamers since no PltB is found for any of the samples in that figure. Therefore, on page 4 line 6 should say (Fig 1e,f) as the support for the lack of heteropentamers.

We have made the suggested change.

2. A further explanation or speculation for the inability to detect PltB following immunoprecipitation and LC/MS/MS as shown in Figure 1d is warranted (in the text and not the figure legend) because PltB is only 4 aa shorter than PltC and is 47% similar according to the supplemental data, and PltC was detected easily.

The ability to detect a protein by LC-MS/MS is often unpredictable, particularly for smaller proteins that offer potentially less peptide targets. Potential protease recognition sites and simply minor variations on the sequence of the generated peptides have a major impact on the ability to generate peptides that can be detected by LC-MS/MS. In any case, the fact is that we simply can't detect this protein by this approach even after generating peptides with different proteases in purified preparations of typhoid toxin. We do have substantial expertise in this approach and we have an excellent Mass Spectrometer in our own laboratory, a Q-Exactive from Thermo Fisher, and therefore this is not a hardware issue. In any case as suggested by the reviewer we have added a simpler statement to the main text to ensure that the reader understands that the LC/MS/MS experiments are not informative with respect to PltB-related interactions.

3. It appears that the A2 moiety of the toxin may have higher affinity for PltC - based on finding more PltC in Fig1d and because when pltC is mutated, more PltB is detected at least in TTIM conditions. Therefore, the question of heterodimers may not be settled until PltB is immunoprecipitated and then PltC is checked for by western blot. However it does appear that the majority of the toxins are in homopentamers.

We have put a substantial effort in the laboratory to generate a suitable anti PltB antibody for IP experiments and even after several attempts we have failed. Furthermore, we have been so far unable to identify a site in PltB that would be permissive for addition of an epitope tag without disrupting its function. However, the PltB blots for the pull downs (both from bacteria grown in medium and from infected cells) clearly show that PltB is readily identified with CdtB pull downs but is completely absent from PltC pull downs. Furthermore, if PltC and PltB did interact this interaction should be much more readily identified than an interaction between PltB and CdtB based on what we know about the structure of AB5 toxins and the way they are assembled. Regardless, however, it is difficult to completely rule out the possibility of a small sub-population of toxins (it would have to be in our calculation less than 5%) with a heteromeric delivery platform. Accordingly, we have softened our language to be less absolute on this matter.

4. There appears to be more PltB immunoprecipitated in the absence of PltC when the cells were grown in the TTIM (Fig 1f) but not the Henle-407 cells (Fig 1e) - or at least much less of a difference, so the statement on p4, line 8 should indicate that the difference is when cells are grown in TTIM, or else the differences should be quantified.

We have added quantification data to demonstrate that when we IP with CdtB, we do indeed see a significant increase in the amount of PltB in strains that lack PltC for bacteria grown in TTIM. This was done to provide further evidence in support of the concept that the two B subunits compete for A subunits. We have adjusted the text to clarify that this phenomenon is demonstrated in the TTIM conditions, as suggested by the reviewer.

5. There appears to be more PltB in the pltC mutant in Figure 1f in both pre and post conditions, although the CdtB-3xF levels appear to be similar. Is this difference meaningful? Does the pltC mutant retain the promoter? The methods section does not describe the nature of the mutations. This information could be added to the extended data.

We have made additions in response to both issues raised by the reviewer. We have added a brief comment concerning the difference in the PltB levels we observed for the whole cell lysates and have speculated that this might be due to protein turnover being slower for the holotoxin compared to the B subunits alone. The pltC deletion is a clean deletion of its coding sequence, start to stop. The pltC locus is at a completely different genome location than the other typhoid toxin genes, so this deletion will not cause polar effects on typhoid toxin expression. We have added a detailed description of the mutations to the methods section.

6. Figure 1f needs more description and labeling. What do the "Pre" and "Post" labels mean (pre- and post-immunoprecipitation, presumably)? What was the top panel probed with, anti-Flag?

We have adjusted the figure legend and the labelling on this panel to improve clarity.

7. p6, second paragraph, line 6 states that *pltC* is expressed in cells in the absence of PhoPQ. Extended Table 1 shows *phoP* and *phoQ* as the 3rd and 4th genes "important for *pltC* expression in infected human cells". So, does that mean that the cut-off for "important" for expression should be above a ratio of 6?

In an effort to keep this article concise, we streamlined the gene regulation aspect of the paper considerably. We can certainly understand the reviewer's confusions/concerns here. Deletions to *phoPQ* do not preclude *pltC* expression during infection. Tabundance of PltC as well as *pltC* reporter expression is essentially identical during infection in WT and *phoPQ* mutant strains (Fig 3C, Extended Data Fig 3 (now Supplementary Fig. 4)). However, while most *phoPQ* mutant bacteria express high levels of *pltC* during infection, this mutation also results in a small subpopulation of bacteria that do not express *pltC* at all (Extended data Fig 4 (now Supp. Fig. 5)). Because such a high proportion of *S. Typhi* express *pltC* during infection and because the genetic screen works on the single bacterium level, the screen is very good at picking up mutants with this phenotype. PhoPQ is known to influence *ssrA/ssrB* expression (the direct regulator of *pltC*) and so it appears that a small subset of *S. Typhi* during infection require PhoPQ stimulation to trigger *SsrAB* activation. To address this reviewers point as well as other issues concerning purines/pyrimidines raised by reviewer #2, we have expanded our discussion of the gene regulation results in the paper to clarify/highlight these issues.

Minor comments:

1. Fig 1d- for the top label in each box, it should be indicated that each of those proteins has a 3xFLAG tag.
2. page 24 line 5 - "dividing" rather than diving.
3. page 25 MOI rather than "moiety of infection"

We have made each of these corrections, as suggested.

Reviewer #2:

The author demonstrate that an "Orphan" B-subunit gene (PltC) in fact assembles into an active toxin that displays a different expression pattern and host cell range from PltB. This is very novel discovery and will be of interest to Salmonella researchers and others.

The work is convincing, with one minor exception. On page 4, while the trend is there, if you want to say PltB and PltC compete for assembly with the A-subunit you would have to do quantitative western blots with multiple independent repeats. If they do not wish to do these experiments, they could just reword this section.

We have added this quantification as suggested - it is included as Supplementary Fig 2. We do indeed observe a statistically significant increase in the amount of PltB that IPs with CdtB (normalized by the amount of CdtB) in *pltC* mutant strains compared to wild-type.

Page 14. Please clarify, "Mean values +/- S.D. are shown for three independent experiments assayed in duplicate (6 total samples)." Did you use N=6 (not correct), or average the duplicates and use N=3 (correct)?

We have corrected this issue, as indicated by the reviewer.

Regarding Figure 3f, pyrimidines and purines seem like two sides of the same coin, I would appreciate some speculation as to why they act as different regulatory signals (purines are also needed for energy?)

As described above in response to a question raised by reviewer #1, we have expanded our discussion of the gene regulation aspect of this paper. Included in this expansion is a discussion of some possible explanations for the purines/pyrimidines finding.

Minor issues for clarity:

1. The text on page 3 uses "sty1364", but on figure 1 it is labeled PltC. On page 4, the big reveal is they are the same. To prevent reader confusion, tentatively call it PltC from the beginning.

We have made this change as suggested by the reviewer.

2. Figure 2a insert. What are the lanes?

We have added this information to the Fig 2 legend.

3. Figure 1a, extended data figure 5, pltC looks blue not purple.

We have changed the hue in these figures to ensure they look purple.

4. Page 5, please define PG.

We have added this definition.

REVIEWERS' COMMENTS:

Reviewer #1 (Remarks to the Author):

The response to the review adequately address the concerns raised in the previous review.

A few small comments: There should be a re-check of the numbering of the tables, which seems to have gotten off. In addition, check lines 180 & 181 for italics, and line 184 needs an "a" inserted before "larger".

REVIEWERS' COMMENTS:

Reviewer #1 (Remarks to the Author):

The response to the review adequately address the concerns raised in the previous review.

A few small comments: There should be a re-check of the numbering of the tables, which seems to have gotten off. In addition, check lines 180 & 181 for italics, and line 184 needs an "a" inserted before "larger".

We thank the reviewer for finding these typos. The table names should now all be correct and we have corrected the italics on lines 180/181 and added "a" to line 184 as suggested.